# EFFICIENT BLOCK CONTRASTIVE LEARNING VIA PARAMETER-FREE META-NODE APPROXIMATION

## ABSTRACT

Contrastive learning has recently achieved remarkable success in many domains including graphs. However contrastive loss, especially for graphs, requires a large number of negative samples which is unscalable and computationally prohibitive with a quadratic time complexity. Sub-sampling is not optimal. Incorrect negative sampling leads to sampling bias. In this work, we propose a meta-node based approximation technique that can (a) proxy *all* negative combinations (b) in quadratic cluster size time complexity, (c) at graph level, not node level, and (d) exploit graph sparsity. By replacing node-pairs with additive cluster-pairs, we compute the negatives in cluster-time at graph level. The resulting Proxy approximated meta-node Contrastive (PamC) loss, based on simple optimized GPU operations, captures the full set of negatives, yet is efficient with a linear time complexity. By avoiding sampling, we effectively eliminate sample bias. We meet the criterion for larger number of samples, thus achieving block-contrastiveness, which is proven to outperform pair-wise losses. We use learnt soft cluster assignments for the meta-node construction, and avoid possible heterophily and noise added during edge creation. Theoretically, we show that real world graphs easily satisfy conditions necessary for our approximation. Empirically, we show promising accuracy gains over state-of-the-art graph clustering on 6 benchmarks. Importantly, we gain substantially in efficiency; up to 2x in training time and over 5x in GPU memory reduction. The code is publicly available.

## 1 INTRODUCTION

Discriminative approaches based on contrastive learning has been outstandingly successful in practice (Guo et al., 2017; Wang & Isola, 2020), achieving state-of-the-art results (Chen et al., 2020a) or at times outperforming even supervised learning (Logeswaran & Lee, 2018; Chen et al., 2020b). Specifically in graph clustering, contrastive learning can outperform traditional convolution and attention-based Graph Neural Networks (GNN) on speed and accuracy (Kulatilleke et al., 2022).

While traditional objective functions encourage similar nodes to be closer in embedding space, their penalties do not guarantee separation of unrelated graph nodes (Zhu et al., 2021a). Differently, many modern graph embedding models (Hamilton et al., 2017; Kulatilleke et al., 2022), use contrastive objectives. These encourage representation of positive pairs to be similar, while making features of the negatives apart in embedding space (Wang & Isola, 2020). A typical deep model consists of a trainable encoder that generates positive and negative node embedding for the contrastive loss (Zhu et al., 2021a). It has been shown that convolution is computationally expensive and may not be necessary for representation learning (Chen et al., 2020a). As the requirement for contrastive loss is simply an encoder, recently researchers have been able to produce state-of-the-art results using simpler and more efficient MLP based contrastive loss implementations (Hu et al., 2021; Kulatilleke et al., 2022). Thus, there is a rapidly expanding interest and scope for contrastive loss based models.

We consider the following specific but popular (Hu et al., 2021; Kulatilleke et al., 2022) form of contrastive loss where $\tau$ is the temperature parameter, $\gamma_{ij}$ is the relationship between nodes $i, j$ and the loss for the $i^{th}$ node is:

$$\ell_i = -\log \frac{\sum_{j=1}^{B} \mathbf{1}_{[j \neq i]} \gamma_{ij} \cdot \exp\left(\text{sim}\left(\boldsymbol{z}_i, \boldsymbol{z}_j\right) \cdot \tau\right)}{\sum_{k=1}^{B} \mathbf{1}_{[k \neq i]} \exp\left(\text{sim}\left(\boldsymbol{z}_i, \boldsymbol{z}_k\right) \cdot \tau\right)}, \tag{1}$$

When no labels are present, sampling of positive and negative nodes plays a crucial role (Kipf & Welling, 2016) and is a key implementation detail in contrastive methods (Velickovic et al., 2019). Positive samples in graphs are typically connected by edges (Kulatilleke et al., 2021), similar to words in a sentence in language modelling (Logeswaran & Lee, 2018). Often data augmentation is used to generate positive samples; Chen et al. (2020b) used crop, coloring, blurring. However, it is harder to obtain negative samples. With no access to labels, negative counterparts are typically obtained via uniform sampling (Park et al., 2022), via synthesizing/augmenting (Chen et al., 2020b) or adding noise. Also, in graphs, adjacency information can be exploited to derive negatives (Hu et al., 2021; Kulatilleke et al., 2022) for feature contrastion. However, while graphs particularly suited for contrastive learning, to be effective, a large number of negative samples must be used (Wang & Isola, 2020) (e.g., 65536 in He et al. (2020)), along with larger batch sizes and longer training compared to its supervised counterparts (Chen et al., 2020b). Prior work has used data augmentation-based contrastive methods Zhu et al. (2020; 2021b), negative samples using asymmetric structures Thakoor et al. (2021) or avoided negative samples altogether via feature-level decorrelation Zhang et al. (2021b). While Thakoor et al. (2021); Zhang et al. (2021b) address complexity and scalability, as seen in Appendix Table 4, their performance can be further improved.

Unlike other domains, such as vision, negative sample generation brings only limited benefits to graphs (Chuang et al., 2020; Zhu et al., 2021a). To understand this phenomenon, observe the raw embedding of USPS image dataset, in the top row of Figure 7 which looks already clustered. A direct consequence of this is that graphs are more susceptible to sampling bias (Chuang et al., 2020; Zhu et al., 2021a). Thus, graph contrastive learning approaches suffer from insufficient negatives and the complex task of sample generation in addition to $O(N^2)$ time complexity required to contrast every negative node.

However, what contrastive loss exactly does remain largely a mystery (Wang & Isola, 2020). For example, Arora et al. (2019)'s analysis based on the assumption of latent classes provides good theoretical insights, yet their explanation on representation quality dropping with large number of negatives is inconsistent with experimental findings (Chen et al., 2020b). Contrastive loss is seen as maximizing mutual information (MI) between two views. Yet, contradictorily, tighter bound on MI can lead to poor representations (Wang & Isola, 2020).

**Motivation**: Prior work has approximated the task in order to approximate the loss. SwAV (Caron et al., 2020) learns to predict a node prototype code of an augmented view from the other view. GRCCA (Zhang et al., 2021a) maps augmented graphs to prototypes using k-means for alignment. PCL (Li et al., 2020) assigns several prototypes of different granularity to an image enforcing its representation to be more similar to its corresponding prototype. However, all these works use some form of data augmentation which assumes that the task-relevant information is not significantly altered and require computationally expensive operations.

Wang & Isola (2020) identifies alignment and uniformity as key properties of contrastive loss: alignment encourages encoders to assign similar features to similar samples; uniformity encourages a feature distribution that preserves maximal information. It is fair to assume that latent clusters are dissimilar. Even with the rare possibility of two identical cluster centers initially, one will usually change or drift apart. It is intuitive that cluster centers should be uniformly distributed in the hyperspace, similar to nodes, in order to preserve as much information of the data as possible. Uniformly distributing points on a hyperspace is defined as minimizing the total pairwise potential w.r.t. a certain kernel function and is well-studied (Wang & Isola, 2020).

Thus, we are naturally motivated to use the cluster centers as meta-nodes for negative contrastion. By aggregation, all its constituent nodes cab be affected. Thus, we avoid sampling, effectively eliminate sample bias, and also meet the criterion of larger number of samples. Learned soft cluster assignments can avoid possible heterophily and add robustness to noise in edge construction.

In this work, we propose a novel contrastive loss, PamC, which uses paramaterless proxy meta-nodes to approximate negative samples. Our approach indirectly uses the full set of negative samples and yet is efficient with a time complexity of $O(N)$. Not only does PamCGC, based on PamC, outperform or match previous work, but it is also simpler than any prior negative sample generation approach, faster and uses relatively less GPU memory. It can be incorporated to any contrastive learning-based clustering model with minimal modifications, and works with diverse data, as we demonstrate using benchmark datasets from image, text and graph modalities.

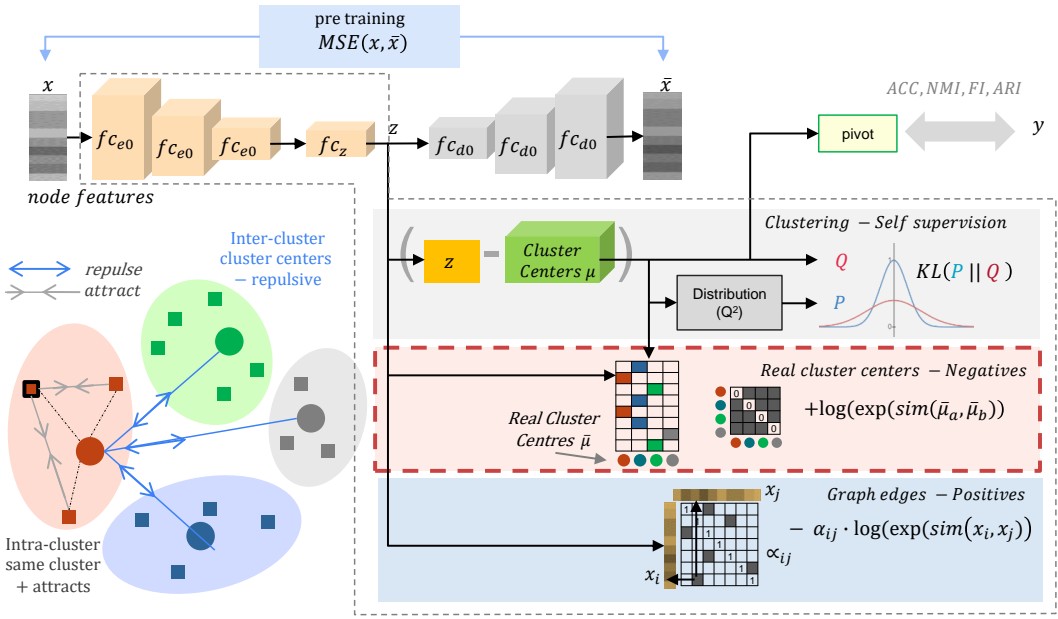

Figure 1: PamCGC jointly learns structure and clustering via probabilistic soft assignment which is used to derive the real cluster centers $\hat{\mu}$, used as proxy for negative samples. Grey dotted section outlines the training components. Cluster centroids $\mu$ are obtained by pre-training an AE for reconstruction. Red dotted section is our core contribution: we use $\hat{\mu}$ as an efficient approximation, computing centroid-pairs instead of node-pairs, achieve block-contrastivness and do so at graph level, not instance level.

To summarize, our main contributions are:

- We introduce an efficient novel parameter-free proxy, PamC, for negative sample approximation that is scalable, computationally efficient and able to include *all* samples. It works with diverse data, *including* graphs. We claim PamC is the first to *implicitly* use the *whole* graph with $O(N)$ time complexity, in addition to further 3-fold gains.

- We provide theoretical proof and show that real world graphs always satisfies the necessary conditions, and that PamCGC is block-contrastive, known to outperform pair-wise losses.

- Extensive experiments on 6 benchmark datasets show PamCGC, using proposed PamC, is on par with or better than state-of-the-art graph clustering methods in accuracy while achieving 2x training time and 5x GPU memory efficiency.

## 2 IMPLEMENTATION

First we describe PamC, which is our parameter-free proxy to efficiently approximate the negative samples, as shown in Figure 1. Next, we introduce PamCGC, a self-supervised model based on PamC to simultaneously learn discriminative embeddings and clusters.

### 2.1 NEGATIVE SAMPLE APPROXIMATION BY META-NODE PROXIES

Contrastive loss makes positive or connected nodes closer and negative or unconnected nodes further away in the feature space (Kulatilleke et al., 2022). However, in order to be effective, *all* negative nodes need to be contrasted with $x_i$ which is computationally expensive. A cluster center is formed by combining all member nodes, and can be seen as an aggregated representation, or a proxy, of its compositional elements. Motivated by this, we use the cluster centers to enforce negative contrastion. Specifically, we contrast every cluster center $\hat{\mu}_i$ with every cluster center $\hat{\mu}_j$ where $i \neq j$.

Following Arora et al. (2019); Chuang et al. (2020), we assume an underlying set of discrete latent classes $C$ which represents semantic content, i.e., similar nodes $x_i, x_j$ are in the same latent class $\hat{\mu}$. Thus, we derive our proxy for negative samples as:

$$\ell_{proxy} = \log \sum_{a=1}^{C} \sum_{b=1}^{C} \mathbf{1}_{[a \neq b]} \exp\left(\text{sim}\left(\hat{\boldsymbol{\mu}}_a, \hat{\boldsymbol{\mu}}_b\right) \cdot \tau\right), \tag{2}$$

Note that, $\ell_{proxy}$ contains no $i$ or $j$ terms! resulting in *three* fold gains. Firstly, we replace $\sum_{i=1}^{N}$, with a more efficient $\sum_{a=1}^{C}$ where $N \gg C$, typically many magnitudes, in almost all datasets, as evident from Table 1. Secondly, the $\ell_{proxy}$ is at *graph level* with time complexity of $O(N)$ rather than an instance level $O(N^2)$. Finally, given real world graphs (especially larger graphs,) are sparse, a sparse implantation for the positives, using edge-lists, will result in a $third$ efficiency gain, which is only possible by not having to operate on the negatives explicitly.

Note that a prerequisite of the proxy approximation is the availability of labels to construct the learned cluster centers $\hat{\mu}$, which we explain in the next section. Thus, the complete graph level contrastive loss can be expressed as:

$$\ell_{Pcontrast} = -\frac{1}{N} \sum_{i=1}^{N} \log \sum_{j=1}^{N} \mathbf{1}_{[j \neq i]} \gamma_{ij} \exp\left(\text{sim}\left(\boldsymbol{z}_i, \boldsymbol{z}_j\right) \cdot \tau\right) + \ell_{proxy}, \tag{3}$$

**Theoretical explanation.** The standard contrastive loss uses Jensen-Shannon divergence, which yields $\log 2$ constant and vanishing gradients for disjoint distributions of positive and negative sampled pairs (Zhu et al., 2021a). However, in the proposed method, positive pairs are necessarily edge-linked (either explicitly or via influence (Kulatilleke et al., 2022)), and unlikely to be disjoint. Using meta-nodes for negatives, which are compositions of multiple nodes, lowers the possibility of disjointness. An algorithm using the average of the positive and negative samples in blocks as a proxy instead of just one point has a strictly better bound due to Jensen's inequality getting tighter and is superior compared to their equivalent of element-wise contrastive counterparts (Arora et al., 2019). The computational and time cost is a direct consequence of node level contrastion. Given, $N \gg clusters$, we circumvent the problem of large $N$ by proposing a proxy-ed negative contrastive objective that operates directly at the cluster level.

Establishing mathematical guarantee: Assume node embeddings $Z = \{z_1, z_2, z_3 \ldots z_N\}$, clusters $\mu = \{\mu_1, \mu_2 \ldots \mu_C\}$, a label assignment operator $\text{label}(z_i)$ such that $\mu_a = \sum_{i=1}^{N} \mathbf{1}_{[i \in \text{label}(z_i)=a]} \cdot z_i$, a temperature hyperparameter $\tau$ and,

$$\text{similarity}(i, j, z_i, z_j) = \text{sim}(z_i, z_j) \begin{cases} 0, & i = j \\ \frac{z_i \cdot z_j}{\|z_i\|\|z_j\|}, & i \neq j \end{cases} \tag{4}$$

Using $\text{sim}(z_i, z_j)$ as the shorthand notation for $\text{similarity}(i, j, z_i, z_j)$, the classic contrastive loss is:

$$loss_{NN} = \frac{1}{N} \sum_{i=1}^{N} \log \left[ \sum_{j=1}^{N} \exp(\text{sim}(i, j, z_i, z_j)\tau) \right], \tag{5}$$

Similarly, we can express the cluster based contrastive loss as:

$$loss_{CC} = \frac{1}{C} \sum_{a=1}^{C} \log \left[ \sum_{b=1}^{M} \exp(\text{sim}(a, b, \mu_a, \mu_b)\tau) \right] \tag{6}$$

As $0 \leq \text{sim} \leq 1.0$, we establish the condition for our inequality as;

$$\frac{loss_{NN}}{loss_{CC}} > \frac{\log(N)}{\log\left[1 + (C-1)e^{\tau}\right]} \tag{7}$$

We provide the full derivation in Appendix A.1.

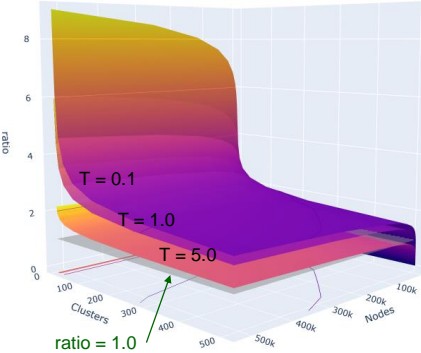

Figure 2: Nodes $N$ vs Clusters $C$ with different $\tau$ temperature values. Grey surface shows the $ratio = 1.0$ inequality boundary. Generally, real world graphs satisfy the condition $ratio > 1.0$ easily. Best viewed in color.

As $C > 1$ (minimum 2 are needed for a cluster), and $log(x) : x > 0$ is strictly increasing, $N > 1 + (C - 1)e^\tau$ is the necessary condition, which is easily satisfied for nearly all real world datasets and as seen in Figure 2 for multiple $\tau$ temperatures.

Thus, as $loss_{NN} > loss_{CC}$, $loss_{NN}$ upper bounds $loss_{CC}$, the more efficient variant. Additionally $loss_{CC}$ benefits from block-contrastiveness (Arora et al., 2019), achieves a lower minima and uses the *fullest possible* negative information. We also show, experimentally, that minimizing $loss_{CC}$ results in effective, and sometimes better, representations for downstream tasks.

## 2.2 CONSTRUCTING THE META-NODE CLUSTER CENTERS ($\hat{\mu}$)

In order to derive the real cluster centers $\hat{\mu}$, which is distinct from the learnt cluster centers $\mu$, we simply aggregate all the node embedding $z$ of a cluster using its label. Even with unlabeled data, label() can be accomplished using predicted soft labels. The intuition here is that, during back-propagation, the optimization process will update the constituent node embeddings, $z$, to incorporate negative distancing. Thus,

$$\hat{\mu}_c = \frac{1}{N} \sum_{i=1}^{N} \mathbf{1}_{[i \in \text{label}(c)]} z_i, \tag{8}$$

where label(c) is either the ground-truth or learnt soft labels. Accordingly, our proxy can equally be used in supervised and unsupervised scenarios and has a wider general applicability as an improvement of the contrastive loss at large. Finlay, Equation 8 can be realized with $softmax()$ and $mean()$ operations, which are well optimized GPU primitives in any machine learning framework. We provide a reference pytorch implementation.

## 2.3 OBTAINING THE SOFT LABELS

Graph clustering is essentially unsupervised. To this end, following Xie et al. (2016); Guo et al. (2017); Wang et al. (2019); Kulatilleke et al. (2022), we use probability distribution derived soft-labels and a self-supervision mechanism for cluster enhancement. Specifically, we obtain soft cluster assignments probabilities $q_{iu}$ for embedding $z_i$ and cluster center $\mu_u$. In order to handle differently scaled clusters and be computationally convenient (Wang et al., 2019), we use the student's $t$-distribution (Maaten & Hinton, 2008) as a kernel for the similarity measurement between the embedding and centroid:

$$q_{iu} = \frac{(1 + \|z_i - \mu_u\|^2 / \eta)^{-\frac{\eta+1}{2}}}{\sum_{u'} (1 + \|z_i - \mu_{u'}\|^2 / \eta)^{-\frac{\eta+1}{2}}}, \tag{9}$$

where, $\eta$ is the Student's $t$-distribution's degree of freedom. Cluster centers $\mu$ are initialized by $K$-means on embeddings from the pre-trained AE. We use $Q = [q_{iu}]$ as the distribution of the cluster assignments of all samples, and $\eta$=1 for all experiments following Bo et al. (2020); Peng et al. (2021); Kulatilleke et al. (2022)

Table 1: Statistics of the datasets (left) and PamCGC hyperparameters (right).

| Dataset | Type | Nodes | Classes | dimension | $\alpha$ | $\beta$ | K | $\tau$ | LR |
|---------|------|-------|---------|-----------|----------|---------|---|--------|-----|
| USPS | Image | 9298 | 10 | 256 | 2 | 2 | 4 | 0.5 | $10^{-3}$ |
| HHAR | Record | 10299 | 6 | 561 | 0.5 | 12.5 | 2 | 1.5 | $10^{-3}$ |
| REUT | Text | 10000 | 4 | 2000 | 1 | 0.2 | 1 | 0.25 | $10^{-4}$ |
| ACM | Graph | 3025 | 3 | 1870 | 0.5 | 0.5 | 1 | 0.5 | $10^{-3}$ |
| CITE | Graph | 3327 | 6 | 3703 | 2 | 2 | 1 | 1 | $10^{-3}$ |
| DBLP | Graph | 4057 | 4 | 334 | 2 | 2.5 | 3 | 0.5 | $10^{-3}$ |

Nodes closer in embedding space to a cluster center has higher soft assignment probabilities in $Q$. A target distribution $P$ that emphasizes the *confident* assignments is obtained by squaring and normalizing $Q$, given by :

$$p_{iu} = \frac{q_{iu}^2 / \sum_i q_{iu}}{\sum_k (q_{ik}^2 / \sum_i q_{ik})}, \tag{10}$$

where $\sum_i q_{iu}$ is the soft cluster frequency of centroid $u$.

Following Kulatilleke et al. (2022), we minimize the KL divergence between $Q$ and $P$ distributions, which forces the current distribution $Q$ to approach the more confident target distribution $P$. KL divergence updates models more gently and lessens severe disturbances on the embeddings (Bo et al., 2020). Further, it can accommodate both the structural and feature optimization targets of PamCGC. We self-supervise cluster assignments [1] by using distribution $Q$ to target distribution $P$, which then supervises the distribution $Q$ in turn by minimizing the KL divergence as:

$$loss_{cluster} = KL(P||Q) = \sum_i \sum_u p_{iu} log \frac{p_{iu}}{q_{iu}}, \tag{11}$$

**The final proposed model,** after incorporating PamC contrastive objective with self-supervised clustering, where $\alpha > 0$ controls structure incorporation and $\beta > 0$ controls cluster optimization is:

$$\text{PamCGC}: \quad \text{L}_{\text{final}} = \alpha \ell_{\text{Pcontrast}}(\text{K}, \tau) + \beta \text{loss}_{\text{cluster}}, \tag{12}$$

## 2.4 COMPLEXITY ANALYSIS

Given input data dimension $d$ and AE layer dimensions of $d_1, d_2, \cdots, d_L$, following Kulatilleke et al. (2022), $O_{AE} = \mathcal{O}(Nd^2 d_1^2 ... d_{L/2}^2)$ for PamCGC-AE. Assuming $K$ clusters, from Equation 9, the time complexity is $O_{cluster} = \mathcal{O}(NK + N \log N)$ following Xie et al. (2016).

For PamC, we only compute $\|z\|_2^2$ and $z_i \cdot z_j$ for the actual positive edges $E$ using sparse matrix resulting in a time complexity $O_+ = \mathcal{O}(NEd_z)$, linear with the number of edges $E$, with $d_z$ embedding dimension. For the negatives, we use the meta-node based negatives $O_- = \mathcal{O}(CC)$ where $C$ is the meta-node. Note that, for real graphs, $N \gg C$ in many magnitudes. Thus, the overall time complexity is linearly related to the number of samples and edges.

## 3 EXPERIMENTS

We evaluate PamCGC on transductive node clustering comparing to state-of-the-art self-supervised, contrastive and (semi-)supervised methods.

**Datasets.** Following Bo et al. (2020); Peng et al. (2021); Kulatilleke et al. (2022), experiments are conducted on six common clustering benchmarks, which includes one image dataset (USPS (Le Cun et al., 1990)), one sensor data dataset (HHAR (Stisen et al., 2015)), one text dataset (REUT (Lewis et al., 2004)) and three citation graphs (ACM[2], CITE[4], and DBLP[3]). For the non-graph data, we use

---

[1] We follow Bo et al. (2020) use of the term 'self-supervised' to be consistent with the GCN training method.

undirected $k$-nearest neighbour (KNN (Altman, 1992)) to generate adjacency matrix $\mathbf{A}$ following Bo et al. (2020); Peng et al. (2021). Table 1 summarizes the datasets.

**Baseline Methods.** We compare with multiple models. K-means (Hartigan & Wong, 1979) is a classical clustering method using raw data. AE (Hinton & Salakhutdinov, 2006) applies K-means to deep representations learned by an auto-encoder. DEC (Xie et al., 2016) clusters data in a jointly optimized feature space. IDEC (Guo et al., 2017) enhances DEC by adding KL divergence-based reconstruction loss. Following models exploit graph structures during clustering: SVD (Golub & Reinsch, 1971) applies singular value decomposition to the adjacency matrix. DGI (Velickovic et al., 2019) learns embeddings by maximizing node MI with the graph. GAE (Kipf & Welling, 2016) combines convolution with the AE. ARGA (Pan et al., 2018) uses an adversarial regularizer to guide the embeddings learning. Following deep graph clustering jointly optimize embeddings and graph clustering: DAEGC (Wang et al., 2019), uses an attentional neighbor-wise strategy and clustering loss. SDCN (Bo et al., 2020), couples DEC and GCN via a fixed delivery operator and uses feature reconstruction. AGCN (Peng et al., 2021), extends SDCN by adding an attention-based delivery operator and uses multi scale information for cluster prediction. CGC (Park et al., 2022) uses a multi-level, hierarchy based contrastive loss. SCGC and SCGC* (Kulatilleke et al., 2022) uses block contrastive loss with an AE and MLP respectively. The only difference between SCGC* and PamCGC is the novel PamC loss, Also as SCGC* is the current state-of-the-art. Thus, it is used as the benchmark.

**Evaluation Metrics.** Following Bo et al. (2020); Peng et al. (2021), we use Accuracy (ACC), Normalized Mutual Information (NMI), Average Rand Index (ARI), and macro F1-score (F1) for evaluation. For each, larger values imply better clustering.

## 3.1 IMPLEMENTATION

The positive component of our loss only requires the actual connections and can be efficiently represented by sparse matrices. Further, the negative component of the loss is graph-based, and not instance based, thus needs to be computed only once per epoch. Thus, by decoupling the negatives, our loss is inherently capable of batching and is trivially parallelizable. Computation of the negative proxy, which is only $C \cdot C$ does not even require a GPU!

For fair comparison, we use the same $500 - 500 - 2000 - 10$ AE dimensions as in Guo et al. (2017); Bo et al. (2020); Peng et al. (2021); Kulatilleke et al. (2022) and the same pre-training procedure, i.e. 30 epochs; learning rate of $10^{-3}$ for USPS, HHAR, ACM, DBLP and $10^{-4}$ for REUT and CITE; batch size of 256. We made use of the publicly available pre-trained AE from Bo et al. (2020). We use a once computed edge-list for training, which is not needed during inference. For training, for each dataset, we initialize the cluster centers from $K$-means and repeat the experiments 10 times with 200 epochs to prevent extreme cases. We cite published accuracy results from Bo et al. (2020); Peng et al. (2021); Kulatilleke et al. (2022) for other models.

For all timing and memory experiments, we replicate the exact same training loops, including internal evaluation metric calls, when measuring performance for fair comparison. Our code will be made publicly available.

## 3.2 RESULTS

We show our hyperparameters in Table 1. Comparison of results with state-of-the-art graph and non-graph datasets are in Table 2 and Table 3, respectively. For the graph data, PamCGC is state-of-the-art for DBLP. A paired-t test shows ACM and CITE results to be best for both SCGC* and PamCGC. In non-graph results, PamCGC comes second best in USPS image data. While results for HHAR are somewhat lagging, PamCGC is the best for REUT. Generally we achieve better results on the natural graph datasets; ACM, DBLP and CITE, while being competitive on other modalities. We present the qualitative results in Appendix A.4.

---

[2] http://dl.acm.org/

[3] https://dblp.uni-trier.de

[4] http://citeseerx.ist.psu.edu/index

Table 2: Clustering performance the three graph datasets (mean±std). Best results are **bold**. Results reproduced from Bo et al. (2020); Peng et al. (2021); Kulatilleke et al. (2022); Park et al. (2022). SCGC (Kulatilleke et al., 2022) uses neighbor based contrastive loss with AE while SCGC* variant uses $r$-hop cumulative Influence contrastive loss with MLP, same as our PamCGC

| Method | DBLP | | | | ACM | | | | CITE | | | |
|---|---|---|---|---|---|---|---|---|---|---|---|---|
| | ACC | NMI | ARI | F1 | ACC | NMI | ARI | F1 | ACC | NMI | ARI | F1 |
| K-means | 38.7±0.7 | 11.5±0.4 | 7.0±0.4 | 31.9±0.3 | 67.3±0.7 | 32.4±0.5 | 30.6±0.7 | 67.6±0.7 | 39.3±3.2 | 16.9±3.2 | 13.4±3.0 | 36.1±3.5 |
| AE | 51.4±0.4 | 25.4±0.2 | 12.2±0.4 | 52.5±0.4 | 81.8±0.1 | 49.3±0.2 | 54.6±0.2 | 82.0±0.1 | 57.1±0.1 | 27.6±0.1 | 29.3±0.1 | 53.8±0.1 |
| DEC | 58.2±0.6 | 29.5±0.3 | 23.9±0.4 | 59.4±0.5 | 84.3±0.8 | 54.5±1.5 | 60.6±1.9 | 84.5±0.7 | 55.9±0.2 | 28.3±0.3 | 28.1±0.4 | 52.6±0.2 |
| IDEC | 60.3±0.6 | 31.2±0.5 | 25.4±0.6 | 61.3±0.6 | 85.1±0.5 | 56.6±1.2 | 62.2±1.5 | 85.1±0.5 | 60.5±1.4 | 27.2±2.4 | 25.7±2.7 | 61.6±1.4 |
| SVD | 29.3±0.4 | 0.1±0.0 | 0.0±0.1 | 13.3±2.2 | 39.9±5.8 | 3.8±4.3 | 3.1±4.2 | 30.1±8.2 | 24.1±1.2 | 5.7±1.5 | 0.1±0.3 | 11.4±1.7 |
| DGI | 32.5±2.4 | 3.7±1.8 | 1.7±0.9 | 29.3±3.3 | 88.0±1.1 | 63.0±1.9 | 67.7±2.5 | 88.0±1.0 | 64.1±1.3 | 38.8±1.2 | 38.1±1.9 | 60.4±0.9 |
| GAE | 61.2±1.2 | 30.8±0.9 | 22.0±1.4 | 61.4±2.2 | 84.5±1.4 | 55.4±1.9 | 59.5±3.1 | 84.7±1.3 | 61.4±0.8 | 34.6±0.7 | 33.6±1.2 | 57.4±0.8 |
| VGAE | 58.6±0.1 | 26.9±0.1 | 17.9±0.1 | 58.7±0.1 | 84.1±0.2 | 53.2±0.5 | 57.7±0.7 | 84.2±0.2 | 61.0±0.4 | 32.7±0.3 | 33.1±0.5 | 57.7±0.5 |
| ARGA | 61.6±1.0 | 26.8±1.0 | 22.7±0.3 | 61.8±0.9 | 86.1±1.2 | 55.7±1.4 | 62.9±2.1 | 86.1±1.2 | 56.9±0.7 | 34.5±0.8 | 33.4±1.5 | 54.8±0.8 |
| DAEGC | 62.1±0.5 | 32.5±0.5 | 21.0±0.5 | 61.8±0.7 | 86.9±2.8 | 56.2±4.2 | 59.4±3.9 | 87.1±2.8 | 64.5±1.4 | 36.4±0.9 | 37.8±1.2 | 62.2±1.3 |
| CGC | 77.6±0.5 | 46.1±0.6 | 49.7±1.1 | 77.2±0.4 | 92.3±0.7 | 72.9±0.7 | 78.4±0.6 | 92.3±0.3 | 69.6±0.6 | 44.6±0.6 | 46.0±0.6 | 65.5±0.7 |
| SDCN | 68.1±1.8 | 39.5±1.3 | 39.2±2.0 | 67.7±1.5 | 90.5±0.2 | 68.3±0.3 | 73.9±0.4 | 90.4±0.2 | 66.0±0.3 | 38.7±0.3 | 40.2±0.4 | 63.6±0.2 |
| AGCN | 73.3±0.4 | 39.7±0.4 | 42.5±0.3 | 72.8±0.6 | 90.6±0.2 | 68.4±0.5 | 74.2±0.4 | 90.6±0.2 | 68.8±0.2 | 41.5±0.3 | 43.8±0.3 | 62.4±0.2 |
| SCGC | 77.7±0.1 | 47.1±0.2 | 51.2±0.2 | 77.3±0.1 | 92.6±0.0 | 73.3±0.0 | 79.2±0.0 | 92.5±0.0 | 73.2±0.1 | 46.8±0.1 | 50.0±0.1 | 63.3±0.0 |
| SCGC* | 77.7±0.1 | 47.1±0.1 | 50.2±0.1 | 77.5±0.1 | **92.6±0.0** | **73.7±0.1** | **79.4±0.1** | **92.6±0.0** | 73.3±0.0 | 46.9±0.0 | **50.2±0.0** | **63.4±0.0** |
| PamCGC | **79.6±0.0** | **49.2±0.1** | **54.7±0.1** | **79.0±0.1** | 92.5±0.0 | 73.7±0.1 | 79.2±0.1 | 92.5±0.0 | **73.3±0.2** | **47.3±0.3** | 50.1±0.4 | **63.4±0.2** |

Table 3: Clustering performance the three non-graph datasets (mean±std). Best results are **bold**; second best is underlined. Results reproduced from Bo et al. (2020); Peng et al. (2021); Kulatilleke et al. (2022). SCGC (Kulatilleke et al., 2022) uses neighbour based contrastive loss with AE while SCGC* variant uses $r$-hop cumulative Influence contrastive loss with MLP, same as our PamCGC

| Dataset | Metric | $K$-means | GAE | VGAE | DAEGC | SDCN | AGCN | SCGC | SCGC* | PamCGC |
|---|---|---|---|---|---|---|---|---|---|---|
| USPS | ACC | 66.82±0.04 | 63.10±0.33 | 56.19±0.72 | 73.55±0.40 | 78.08±0.19 | 80.98±0.28 | 82.90±0.08 | **84.91±0.06** | 84.20±0.24 |
| | NMI | 62.63±0.05 | 60.69±0.58 | 51.08±0.37 | 71.12±0.24 | 79.51±0.27 | 79.64±0.32 | 82.51±0.07 | **84.16±0.10** | 80.32±0.38 |
| | ARI | 54.55±0.06 | 50.30±0.55 | 40.96±0.59 | 63.33±0.34 | 71.84±0.24 | 73.61±0.43 | 76.48±0.11 | **79.50±0.06** | 77.75±0.56 |
| | F1 | 64.78±0.03 | 61.84±0.43 | 53.63±1.05 | 72.45±0.49 | 76.98±0.18 | 77.61±0.38 | 80.06±0.05 | **81.54±0.06** | 78.82±0.17 |
| HHAR | ACC | 59.98±0.02 | 62.33±1.01 | 71.30±0.36 | 76.51±2.19 | 84.26±0.17 | 88.11±0.43 | **89.49±0.22** | 89.36±0.16 | 84.94±1.09 |
| | NMI | 58.86±0.01 | 55.06±1.39 | 62.95±0.36 | 69.10±2.28 | 79.90±0.09 | 82.44±0.62 | 84.24±0.29 | **84.50±0.41** | 79.54±0.65 |
| | ARI | 46.09±0.02 | 42.63±1.63 | 51.47±0.73 | 60.38±2.15 | 72.84±0.09 | 77.07±0.66 | **79.28±0.28** | 79.11±0.18 | 72.57±1.20 |
| | F1 | 58.33±0.03 | 62.64±0.97 | 71.55±0.29 | 76.89±2.18 | 82.58±0.08 | 88.00±0.53 | **89.59±0.23** | 89.48±0.17 | 84.13±1.30 |
| REUT | ACC | 54.04±0.01 | 54.40±0.27 | 60.85±0.23 | 65.50±0.13 | *79.30±0.11* | 79.30±1.07 | **80.32±0.04** | 79.35±0.00 | **81.78±0.01** |
| | NMI | 41.54±0.51 | 25.92±0.41 | 25.51±0.22 | 30.55±0.29 | *56.89±0.27* | 57.83±1.01 | 55.63±0.05 | 55.16±0.01 | **59.13±0.00** |
| | ARI | 27.95±0.38 | 19.61±0.22 | 26.18±0.36 | 31.12±0.18 | *59.58±0.32* | 60.55±1.78 | 59.67±0.11 | 57.80±0.01 | **63.51±0.03** |
| | F1 | 41.28±2.43 | 43.53±0.42 | 57.14±0.17 | 61.82±0.13 | *66.15±0.15* | 66.16±0.64 | 63.66±0.03 | 66.54±0.01 | **69.48±0.03** |

## 3.3 PERFORMANCE

In Figure 3 we compare the GPU based training time and GPU memory. Our model times also include the time taken for the cumulative influence computation. For all the datasets, PamCGC is superior by 2.2x training time and 5.3x GPU memory savings. Especially, for larger datasets USPS, HHAR and REUT, PamCGC uses 5.2,7.7,8.7x less GPU memory.

Additionally, we used CITE dataset (3327 nodes) to create synthetics nodes. For a scale factor $n$, as contact nodes $n$ times, along with edge-lists. Figure 3(right) shows the scaled edges and nodes for scale factors $5, 10, 15 \cdots 45$ and the GPU memory and training time for 1 epoch on Google colab T4 GPU with 16GB memory. Without PamC, scales over 5 is not possible due to running out of memory. With PamC over x45 (150,000 nodes) is possible. GPU and memory increase is liner confirming the theoretical time complexity. We used CITE as it is a very common dataset. We used synthetic node creation to capture variation over node size. Appendix A.8 shows GPU time breakup. Appendix A.6 shows the CITE dataset results with PamC when scaled from $1 \ldots 20$ in steps of 1.

## 3.4 ABLATION STUDY

To investigate PamCs ability to generalize to other models, we incorporate it to SDCN and AGCN models, modified for contrastive loss. Figure 4 shows the GPU training time and accuracy. As PamC is a loss function, there is no difference in the inference times. As expected, training times

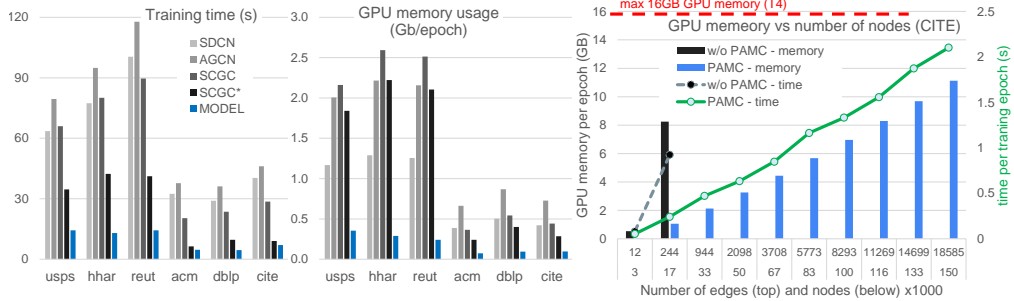

Figure 3: GPU performance from the pytorch profiler on Google Colab with T4 16Gb GPU. **left**:training time for 200 epochs. **center**:memory utilization per epoch. **right**:graph size vs time and memory on synthetic CITE data per epoch; W/o PamC, model runs out of memory after 17,000 nodes. With PamC, 150,000 nodes and over 18 million edges can be handled on the T4's 16GB. Note that SCGC* only differs from PamCGC by its use of the novel proxy-ed PamC to which we solely attribute the time and memory savings.

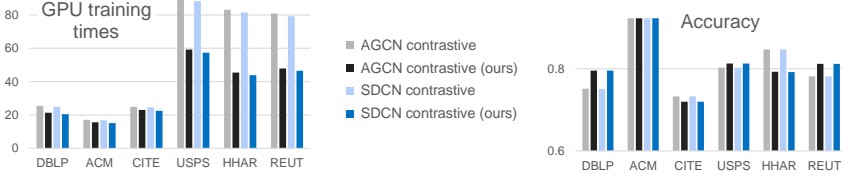

Figure 4: Left:GPU training times with PamCGC on SDCN and AGCN is consistently lower, and significant in large datasets (usps,hhar,reut). Right:Accuracy loss of the approximation is very low. For dblp, usps, reut accuracy is actually better.

are significantly shorter, with (often better) training accuracy due to block contrastiveness. Note that PamC only improves loss computation efficiency. Majority of the SDCN and AGCN computation time is spent in their GNNs convolution operations.

We also carry out extensive experimentation to assess the behavior of hyperparameters. PamC is robust to changes in hyperparameter values and performs best with a learning rate of $0.001$, as shown in Appendix A.2. Further, PamC accuracy against all hyperparameter combinations is generally equal or better than the less efficient non proxy-ed contrastive loss variant, as seen in Appendix A.3.

## 3.5 FUTURE WORK

Our parameter-free proxy-ed contrastive loss uses the full positive edge information which, as some of our experiments has shown, is redundant. For example, USPS gives similar results with 40% positive edges removed. An algorithm to drop un-informative edges may result in further efficiency improvements, which we leave for future work. While theoretically possible, it would be interesting to see how our proxy-ed contrastive loss works with semi or fully supervised data. Further study is needed to explore how hard cluster centers effect the optimization process.

## 4 CONCLUSION

In this work, we present an efficient parameter-free proxy approximation to incorporate negative samples in contrastive loss for joint clustering and representation learning. We eliminate sample bias, achieve block contrastiveness and $0(N)$. Our work is supported by theoretical proof and empirical results. We improve considerably over previous methods accuracy, speed and memory usage. Our approach differs from prior self-supervised clustering by the proxy mechanism we use to incorporate **all** negative samples efficiently. The strength of this simple approach indicates that, despite the increased interest in graphs, effective contrastive learning remains relatively unexplored.

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

# A  APPENDIX

## A.1  PROOFS OF THEORETICAL RESULTS - DERIVATION OF EQUATION 7

Assume node embeddings $Z = \{z_1, z_2, z_3 \dots z_N\}$, clusters $\mu = \{\mu_1, \mu_2 \dots \mu_C\}$, a label assignment operator $\text{label}(z_i)$ such that $\mu_a = \sum_{i=1}^{N} \mathbf{1}_{[i \in \text{label}(z_i)=a]} \cdot z_i$, a hyperparameter $\tau$ related to the temperature in contrastive loss and

$$\text{similarity}(i, j, z_i, z_j) = \text{sim}(z_i, z_j) \begin{cases} 0, & i = j \\ \frac{z_i \cdot z_j}{\|z_i\| \|z_j\|}, & i \neq j \end{cases} \tag{13}$$

We use $\text{sim}(z_i, z_j)$ as the shorthand notation for $\text{similarity}(i, j, z_i, z_j)$ interchangeably for brevity.

We begin with Equation 1, which is the popular form of contrastive loss (Hu et al., 2021; Kulatilleke et al., 2022). With $\tau$ as the temperature parameter, $\gamma_{ij}$ the relationship between nodes $i, j$, the loss for the $i^{th}$ can be expanded as:

$$\ell_i = + \log \sum_{j=1}^{B} \mathbf{1}_{[j \neq i]} \exp\left(\text{sim}\left(\boldsymbol{z}_i, \boldsymbol{z}_j\right) \tau\right) - \log \sum_{j=1}^{B} \mathbf{1}_{[j \neq i]} \gamma_{ij} \exp\left(\text{sim}\left(\boldsymbol{z}_i, \boldsymbol{z}_j\right) \tau\right), \tag{14}$$

where, the first part on the right corresponds to the negative node contrasting portion and the second portion contrasts the positives for node $i$. From Equation 14, for all nodes $N$, we take to negative node contrasting portion, by averaging over $N$ nodes to obtain:

$$loss_{NN} = \frac{1}{N} \sum_{i=1}^{N} log \left[ \sum_{j=1}^{N} e^{\text{sim}(i,j,z_i,z_j)\tau} \right], \tag{15}$$

Note our use of the more concise $\text{sim}()$ and the compact $e$ notation over $\exp()$ interchangeably for compactness reasons.

We expand Equation 15, together with $e^0 = 1$ in cases where $i = j$, as:

$$
\begin{aligned}
loss_{NN} = \frac{1}{N} \Big[ & log \Big( \quad 1 \quad + e^{sim(z_1,z_2)\tau} + e^{sim(z_1,z_3)\tau} + e^{sim(z_1,z_4)\tau} \dots + e^{sim(z_1,z_N)\tau} \Big) + \\
& log \Big( e^{sim(z_2,z_1)\tau} + \quad 1 \quad + e^{sim(z_2,z_3)\tau} + e^{sim(z_2,z_4)\tau} \dots + e^{sim(z_2,z_N)\tau} \Big) + \\
& log \Big( e^{sim(z_3,z_1)\tau} + e^{sim(z_3,z_2)\tau} + \quad 1 \quad + e^{sim(z_3,z_4)\tau} \dots + e^{sim(z_3,z_N)\tau} \Big) + \\
& \qquad\qquad\qquad\qquad \dots \\
& log \Big( e^{sim(z_N,z_1)\tau} + e^{sim(z_N,z_2)\tau} + e^{sim(z_N,z_3)\tau} + e^{sim(z_N,z_4)\tau} \dots + \quad 1 \quad \Big) \Big]
\end{aligned}
\tag{16}
$$

Similarly, we can express the cluster based contrastive loss as:

$$loss_{CC} = \frac{1}{C} \sum_{a=1}^{C} log \left[ \sum_{b=1}^{M} e^{\text{sim}(a,b,\mu_a,\mu_b)\tau} \right] \tag{17}$$

with the following expansion:

$$
\begin{aligned}
loss_{CC} = \frac{1}{C} \Bigg[ & log \Big( \quad 1 \quad + e^{sim(\mu_1,\mu_2)\tau} + e^{sim(\mu_1,\mu_3)\tau} + e^{sim(\mu_1,\mu_4)\tau} \ldots + e^{sim(\mu_1,\mu_C)\tau} \Big) + \\
& log \Big( e^{sim(\mu_2,\mu_1)\tau} + \quad 1 \quad + e^{sim(\mu_2,\mu_3)\tau} + e^{sim(\mu_2,\mu_4)\tau} \ldots + e^{sim(\mu_2,\mu_C)\tau} \Big) + \\
& log \Big( e^{sim(\mu_3,\mu_1)\tau} + e^{sim(\mu_3,\mu_2)\tau} + \quad 1 \quad + e^{sim(\mu_3,\mu_4)\tau} \ldots + e^{sim(\mu_3,\mu_C)\tau} \Big) + \\
& \qquad\qquad\qquad\qquad \ldots \\
& log \Big( e^{sim(\mu_C,\mu_1)\tau} + e^{sim(\mu_C,\mu_2)\tau} + e^{sim(\mu_C,\mu_3)\tau} + e^{sim(\mu_C,\mu_4)\tau} \ldots + \quad 1 \quad \Big) \Bigg]
\end{aligned}
\tag{18}
$$

If, $loss_{NN}^{min} > loss_{CC}^{max}$, we have $\frac{loss_{NN}}{loss_{CC}} > 1$. Next we show the conditions necessary for establishing this inequality.

As $0 \leq sim \leq 1.0$, we obtain the $min$ using $sim_{min} = 0$:

$$
\begin{aligned}
loss_{NN}^{min} &= \frac{1}{N} \Bigg[ log \left( 1 + e^0 + e^0 + \ldots + e^0 \right) + \cdots + log \left( 1 + e^0 + e^0 + \ldots + e^0 \right) \Bigg] \\
&= log \left[ 1 + (N-1)e^0 \right] \\
&= log(N)
\end{aligned}
\tag{19}
$$

Similarly, we can obtain the $max$, using $sim_{max} = 1.0$:

$$
\begin{aligned}
loss_{CC}^{max} &= \frac{1}{C} \Bigg[ log \left( 1 + e^{1.\tau} + e^{1.\tau} + \ldots + e^{1.\tau} \right) + \cdots + log \left( 1 + e^{1.\tau} + e^{1.\tau} + \ldots + e^{1.\tau} \right) \Bigg] \\
&= log \left[ 1 + (C-1)e^{\tau} \right]
\end{aligned}
\tag{20}
$$

Combining Equation 19 and Equation 20, we establish the necessary condition for our inequality, Equation 7 as;

$$
\frac{loss_{NN}}{loss_{CC}} > \frac{log(N)}{log \left[ 1 + (C-1)e^{\tau} \right]}
$$

This derivation is used in Section 2.1, where we show how the condition is almost always satisfied in real graphs. As a result, $loss_{NN}$ upper bounds $loss_{CC}$. Note that a lower loss is better.

## A.2 Hyperparameters vs Accuracy

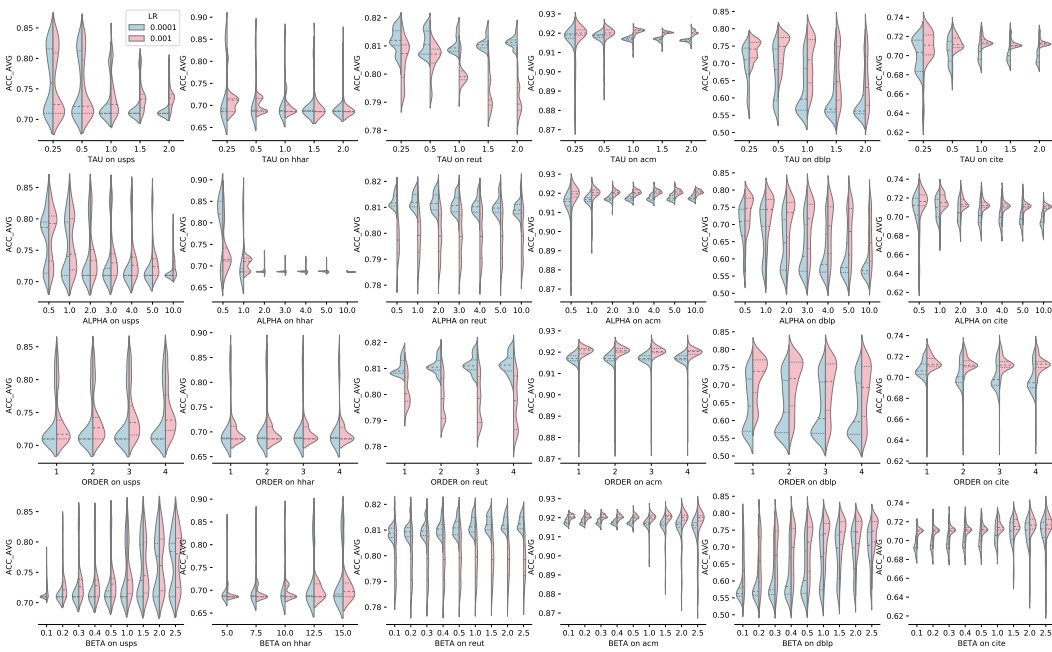

Figure 5: Ablation study on the hyperparameters. TAU=$\tau$, ALPHA=$\alpha$, ORDER=$R$ and LR denotes learning rate. A hyperparameter with higher and more condensed distribution represents its superiority over its counterpart. PamCGC is robust to $\tau, \alpha, R$ and best with a learning rate 0f 0.001. Best viewed in color.

## A.3 Hyperparameter behaviour with and without PamC

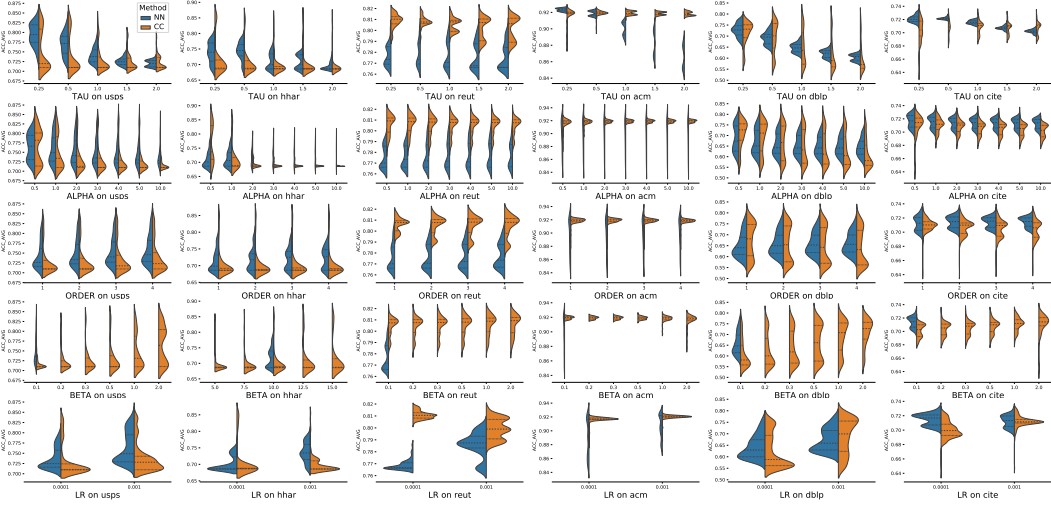

Figure 6: Comparison of hyperparameters with and without PamC. TAU=$\tau$, ALPHA=$\alpha$, ORDER=$R$ and LR denotes learning rate. A hyperparameter with higher and more condensed distribution represents its superiority over its counterpart. PamC is generally better in accuracy for majority of the hyperparameter combinations. Best viewed in color.

### A.4  QUALITATIVE RESULTS

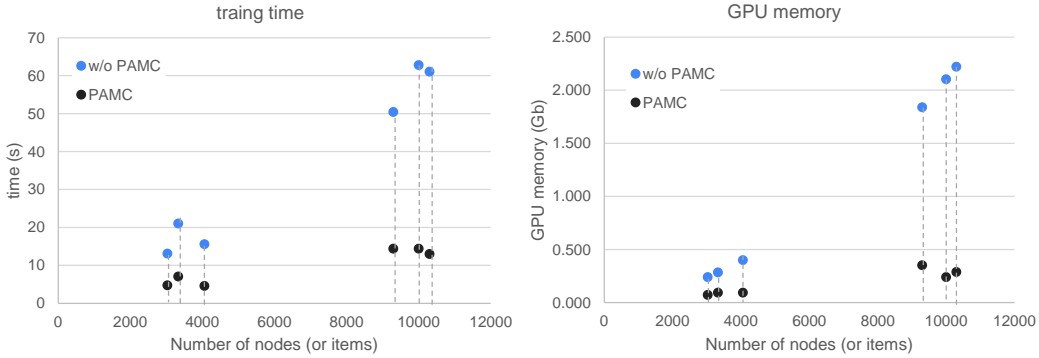

Figure 7: Visual comparison of embeddings; top: raw data, second row: after AE pre-training, third-row: from SCGC*, and last-row: from PamCGC*. Colors represent ground truth groups. Black squares, $\hat{\mu}$, are the approximated meta-nodes. Red dots, $\mu$, are the cluster centroids.

We use UMAP (McInnes et al., 2018), in Figure 7, to get a visual understanding of the raw and learnt embedding spaces. Except for USPS, which is a distinct set of $0 \cdots 9$ handwritten digits (raw 1), we see that all other datasets produce quite indistinguishable clusters. Clustering is nearly non-existent in the (last 3) graph datasets. This clearly shows a characteristic difference in graph data, which can lead to high samplings bias. Note that $\hat{\mu} \neq \mu$ for any meta-node.

### A.5  DATASET SIZE VS GPU MEMORY AND TIME, WITH AND WITHOUT PAMC

Figure 8: Graph size vs GPU memory and training time with and without PamC. Using PamC is generally better and is more effective with larger graph sizes. Best viewed in color.

## A.6 Improved GPU memory and training time on synthetic CITE dataset

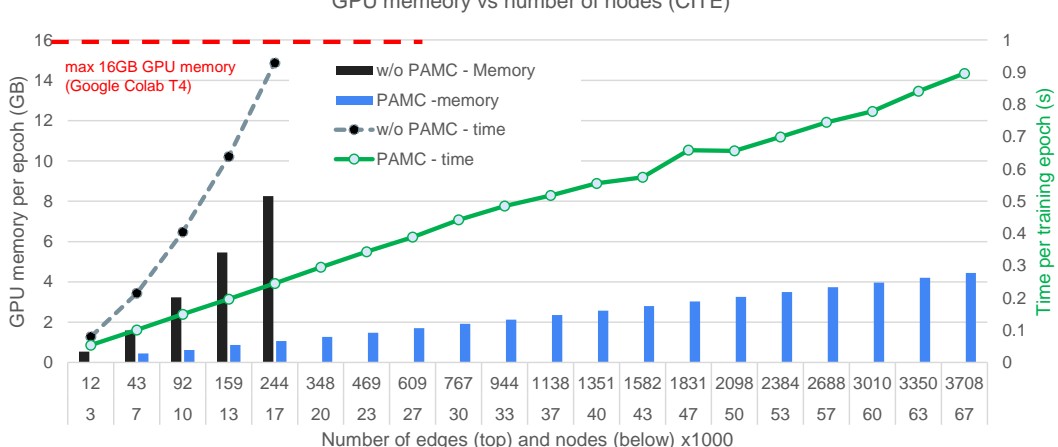

Figure 9: Graph size vs GPU memory and training time with and w/o PamC for synthetic CITE dataset scaled from 1 to 20. Scaled edge and node sizes are indicated in the x-axis. PamC achieves linear time and memory and is more effective with larger graph sizes. W/o PamC, model runs out of memory on Google Colab T4 (16GB GPU memory). Best viewed in color.

## A.7 Results from selected works on the CITE dataset

Table 4: Results for CITE dataset shows PamC is competitiveness in terms of accuracy. ‡Results reproduced from Zheng et al. (2022)

| Model | Accuracy – CITE dataset |
|---|---|
| GRACE Zhu et al. (2020) | 71.7 ± 0.6 ‡ |
| GCA Zhu et al. (2021b) | 71.2 ± 0.2 ‡ |
| BGRL Thakoor et al. (2021) | 71.6 ± 0.4 ‡ |
| CCA-SSG Zhang et al. (2021b) | 73.1 ± 0.3 |
| PamC(Ours) | **73.3 ± 0.2** |

## A.8 GPU workload breakdown with and without PamC.

Table 5: The GPU time breakdown for USPS dataset for 200 epochs on Colab T4 (16GB). The model forward figures (1.272 and 2.257) are different because the GPU is caching the results in the case of no PamC. During inference, these figures are identical.

| Description of task | w/o PamC | With PamC |
|---|---|---|
| Forward pass | 24.378s | 6.670s |
| Model forward (computation of z) | 1.272s | 2.257s |
| KL (self-supervision loss) | 7.200ms | 13.448ms |
| Pseudo label and negative computation | 18.947s | 28.970ms = 0.02897s |
| Contrastive loss (once the pairs are computed) | 2.548s | 1.914s |

