# OpenReview forum: "Efficient block contrastive learning via parameter-free meta-node approximation"
_ICLR.cc/2023/Conference — Submitted to ICLR 2023_

### Official Review · Reviewer_6uyq · 2022-10-23

**Confidence:** 3
**Correctness:** 3
**Technical Novelty And Significance:** 3
**Empirical Novelty And Significance:** 3
**Recommendation:** 6

**Clarity, Quality, Novelty And Reproducibility:**

Clarity: fair, see questions above
Quality: reasonable
Novelty: needs clarification


**Strength And Weaknesses:**

## Strength
1. The direction of more efficient negative sample mining is important for contrastive learning.
2. The idea of utilizing the clustering centers as negative samples saves the sampling cost.
3. Extensive experiments demonstrate comparable performance with baseline methods while achieving good speedup ratio for training time.

## Weaknesses
1. Discussion on related works is not enough, which makes the novelty a little unclear. Is this work the first to utilize averaged embeddings as negative samples for contrastive learning? What are the closest related works?
2. It is not clear how the proposed method can save inference time
3. How long does it take to evaluate the soft label? Is it included in the training time?

**Summary Of The Paper:**

This paper proposes to utilize the meta-node (average of node embeddings within a cluster) for constructing negative samples in contrastive learning. It claims that this approach is more efficient than previous sampling based methods and it leads naturally to block contrastive learning which could be beneficial. Experiments on various datasets demonstrate the performance of the proposed method on different modalities including image, text and graph.

**Summary Of The Review:**

Overall the proposed method is well motivated and achieves reasonable speedup for training. I have some questions about the novelty and experiments. I may consider adjusting my rating after discussing with the authors.

---

> ### Author Response · Authors · 2022-11-14
> **Response**
>
> **Q1. On discussion on related works**
>
> Please see our response to reviewer X8V7 and GN9c. We have added 2 sections (Page 2, end of para 1 and a new para 4 in motivation) based on suggested prior work to address the shortcoming related to background.  These sections in marked in blue.
>
> **Q2. Explanation on why the inference times differ (and a grateful apology)**
> Thank you, we are very grateful for bringing this issue to our notice. PAMC is a loss, so you are correct, inference times should be unaffected, and our reported figures had an error.
>
> We took the timing from pytoch profiler on Colab in August. The performance for PAMC was taken in September. Between the time, Colab has updated internally. Running SCGC* and PAMC now gives identical inference times. Accordingly, we have removed the inference Chart (Figure 3) and replaced it with a scalability chart that shows how the memory and training time varies with graph node and edge sizes, amended reference to inference times and added more graphs to show performance with increasing node and edge sizes using a synthetic version of the CITE dataset.
>
> Also please note that, in Figure 5, where we retrofit PAMC to AGCN and SDCN model, the inference times with and without PAMC are the same. There we state:
> “As PAMC is a loss function, there is no difference in the inference times. As expected, training times are significantly shorter, with (often better) training accuracy due to block contrastiveness. Note that PAMC only improves loss computation efficiency.”
>
> We have now updated the mention of inference times throughout the paper and added the new training time gain, which is x2.2 (different from the previously reported x3). There is no change in memory usage gain, which is x5.
>
> **Q3. On time taken to evaluate the soft label**
>
> Yes, the training times are inclusive of soft label creation times. The bellow table shows the time breakdown for USPS 200 epochs on Colab T4 (16GB). The model forward figures (1.272 and 2.257) are different because the GPU is caching the results in the case of no PAMC. During inference, these figures are identical.
>
> | Description of task                            | w/o PAMC| With PAMC |
> |------------------------------------------------|---------------------|----------------------|
> | Forward pass                                   | 24.378s             | 6.670s               |
> | Model forward (computation of z)               | 1.272s              | 2.257s               |
> | KL (self-supervision loss)                     | 7.200ms             | 13.448ms             |
> | Pseudo label and negative computation          | 18.947s             | 28.970ms = 0.02897s  |
> | Contrastive loss (once the pairs are computed) | 2.548s              | 1.914s               |
>
>
> For added insights we will include the table in Appendix Table 5. This same information can be reproduced from the code already provided in supplementary material.

---

### Official Review · Reviewer_cHFr · 2022-10-24

**Confidence:** 2
**Correctness:** 4
**Technical Novelty And Significance:** 2
**Empirical Novelty And Significance:** Not applicable
**Recommendation:** 3

**Clarity, Quality, Novelty And Reproducibility:**

- The clarity and reproducibility are good.
- The novelty and originality may be not significant.

**Strength And Weaknesses:**

Strength
- It makes sense to replacing the contrastive between negative samples with that between class clusters to reduce the complexity.
- The experimental evaluations are sufficient.
- The writing and organization are good.

Weakness
- My main concern is the novelty may not be significant. Although authors call the cluster center as meta-nodes, they are actually the prototype of the cluster. There are many efforts have be paid on prototypical contrastive learning, such as [1] and its variants, which also takes cluster centers into considerations and reduce the complexity.

[1] Junnan Li, Pan Zhou, Caiming Xiong, Steven C. H. Hoi: Prototypical Contrastive Learning of Unsupervised Representations. ICLR 2021
2020


**Summary Of The Paper:**

This paper extends the contrastive learning on graph by replacing the contrastive between negative samples with that between class clusters with theoretical analysis. It possesses the attractive characteristic of reducing complexity. Experimental evaluations demonstrate its effectiveness and efficiency.

**Summary Of The Review:**

This paper possesses clear motivation, good writing and organization, and sufficient evaluations. My main concern is the novelty, since existing prototypical contrastive learning also reduce complexity by considering cluster centers.

---

> ### Author Response · Authors · 2022-11-14
> **Response**
>
> **On novelty and significance and difference from prototypes of the cluster**
>
> Please see our response to reviewer X8V7 on other works including [1]. We have updated the related works, including [1] in the Motivation section.
>
> Our core novelty is the efficient proxy mechanism for the negative sampling. We show that it can work on graphs, images, text and (sensor) data. We also retrofit this on other state of the art models (AGCN, SDCN) and show that, adding PAMC is sufficient to improve these models.  Thus, our work can be seen as an enhancement to the broader contrastive learning technique, in terms of efficiency, robustness and accuracy, with broader implications.
>
> In our work, we show that it is possible to replace the quadratic negative sampling process (Equation 1, denominator) by an approximation using meta-nodes, 𝜇^, shown in Equation 2.
> We show this approximation has following benefits
>
> 1. Computation of the approximator, 𝜇^, only needs additions (a well optimized GPU primitive)
> 2. No sampling, hence, no sample bias.
> 3. time complexity drops from quadruple node to CxC, independent from node size and computable even on a CPU
> 4. Negative computation is no longer node based. There are no i,j terms on the right hand side. It is computed once per epoch.
> 5. it is block contrastive as we aggregate. Thus, it is robust and well behaved. (Contrastive loss uses Jensen-Shannon divergence, which yields log 2 constant and vanishing gradients for disjoint distributions of positive and negative sampled pairs (Zhu et al., 2021). Aggregations are less likely to be disjoint.
>
> Further, we have added new Figures 3 and 6, showing the time and memory against node size 	using a synthetic graph in the Performance section to show the scalability training and memory benefits of our approach. The foundation of the approximation has been mathematically validated and we provide theoretical complexity estimates which our experiments confirm.
>
> We believe the approximation solution we propose is general (on graphs, images, text) and can be applied on other contrastive learning based **and** non-contrastive methods, to see immediate performance and accuracy gains.

---

### Official Review · Reviewer_X8V7 · 2022-10-25

**Confidence:** 3
**Clarity, Quality, Novelty And Reproducibility:** Please see weaknesses for additional …
**Correctness:** 3
**Technical Novelty And Significance:** 2
**Empirical Novelty And Significance:** 2
**Recommendation:** 5

**Strength And Weaknesses:**

**Strengths**
* The proposed approach seems to be widely empirically effective and is accurate. The approach is very computationally efficient compared to baseline approaches.
* The proposed approach uses an interesting clustering-based objective that seems well motivated.

**Weaknesses**
* Clustering objectives have been studied in other contrastive learning approaches e.g. [1, 2, 3] among a wide array of other approaches in deep clustering. It would greatly improve the paper to consider how these approaches relate to the proposed approach.
* Could the authors clarify the statement
>We provide theoretical proof and show that real world graphs always satisfies the necessary
conditions, and that PamCGC is block-contrastive, known to outperform pair-wise losses
* While the proposed approach offers empirical advantages and is an interesting idea, I am wondering if the methodological depth is sufficient for the ICLR "bar". It would seem that the novelty is the combination of clustering ideas in graph representation learning. I think this is nice, but I feel the authors could clarify how their contribution investigates deeper questions in this area than past work. For instance, efficiency is a major advantage of the proposed approach. Understanding tradeoffs here is in Figure 2. It seems like more emphasis could be placed in this understanding.

[1] Caron, Mathilde, Ishan Misra, Julien Mairal, Priya Goyal, Piotr Bojanowski, and Armand Joulin. "Unsupervised learning of visual features by contrasting cluster assignments." Advances in Neural Information Processing Systems 33 (2020): 9912-9924.

[2] Zhang, Chunyang, Hongyu Yao, C. L. Chen, and Yuena Lin. "Graph Representation Learning via Contrasting Cluster Assignments." arXiv preprint arXiv:2112.07934 (2021).

[3] Li, Junnan, Pan Zhou, Caiming Xiong, and Steven CH Hoi. "Prototypical contrastive learning of unsupervised representations." arXiv preprint arXiv:2005.04966 (2020).

**Summary Of The Paper:**

This paper considers the task of representation learning for graph data. The proposed objective is a contrastive learning objective that is based on using clusters in negative sampling. Empirical analysis compares the proposed approach to a wide variety of representation learning and clustering approaches for graphs.

**Summary Of The Review:**

This paper presents an empirically effective approach for graph representation learning and clustering. The approach, which uses clustering  is more efficient than previous work. The paper could be improved with more complete description of the landscape of related work on representation learning and clustering. The paper could be further improved with more clarity on the technical depth and revealed performance tradeoffs.

---

> ### Author Response · Authors · 2022-11-14
> **Response**
>
> **Q1. On related prototyping approaches**
>
> Thank you for bringing these works to our attention.
>
> Different from PAMC that uses augmentation-free approach and works on graphs, SwAV[1] uses data augmentation for images and addresses online learning. Specifically, it learns to predict a node prototype code of an augmented view from the other view. GRCCA[2] creates 2 augmented graphs, generates 2 representation sets, map them to 2 prototypes using k-means and learns to align the prototypes.  Different from GRCCA, we only use k-means at the start to initialize weights, and our proxy meta-nodes (the counterpart to prototypes in GRCCA) only requires simple aggregation. We found that meta-nodes are better than cluster centres. Figure 4, in our paper, shows the cluster centroids (in red squares) along with the meta-nodes (in black). PCL[3] works on images by assigning several prototypes of different granularity to each image, and enforcing its representation to be more similar to its corresponding prototypes compared to other prototypes.  We compute the proxy as a simple aggregation and each node has a unique meta-node.
>
> We have added the following into the Motivation section: ""
> Prior work has approximated the task in order to approximate the loss. SwAV (Caron et al., 2020) learns to predict a node prototype code of an augmented view from the other view. GRCCA (Zhang et al., 2021a) maps augmented graphs to prototypes using k-means for alignment. PCL (Li et al., 2020) assigns several prototypes of different granularity to an image enforcing its representation to be more similar to its corresponding prototype. However, all these works use some form of data augmentation which assumes that the task-relevant information is not significantly altered and require computationally expensive operations""
>
> **Q2. Clarification of "We provide theoretical proof and show that real world graphs always satisfies the necessary conditions, and that PamCGC is block-contrastive, known to outperform pair-wise losses"**
>
> In our work, we show that it is possible to replace the quadratic negative sampling process (Equation 1, denominator) by an approximation using meta-nodes, 𝜇^, shown in Equation 2.
>
> In order to do so, Equation 7 needs to be satisfied (Please Appendix A for derivation).  In sub section “Establishing mathematical guarantee” we show that almost all real-world graphs can satisfy this condition. Figure 2, in the paper, also shows the full set of node and cluster combinations that satisfies Equation 7.
>
> PAMC it is block contrastive as we aggregate. Thus, it is robust and well behaved. (Contrastive loss uses Jensen-Shannon divergence, which yields log 2 constant and vanishing gradients for disjoint distributions of positive and negative sampled pairs (Zhu et al., 2021). Aggregations are less likely to be disjoint.
>
> **Q3. On novelty over traditional contrastive approach and the use of proxy to improve efficiency**
> Our core novelty is the efficient proxy mechanism for the negative sampling. We show that it can work on graphs, images, text and (sensor) data. We also retrofit this on other state of the art models (AGCN, SDCN) and show that, adding PAMC is sufficient to improve these models.  Thus, our work can be seen as an enhancement to the broader contrastive learning technique, in terms of efficiency, robustness and accuracy, with broader implications.
> We show this approximation has following benefits
> 1. Computation of the approximator, 𝜇^, only needs additions (a well optimized GPU primitive)
> 2. No sampling, hence, no sample bias.
> 3. time complexity drops from quadruple node to CxC, independent from node size and computable even on a CPU
> 4. Negative computation is no longer node based. There are no i,j terms on the right hand side. It is computed once per epoch.
> 5. it is block contrastive as we aggregate. Thus, it is robust and well behaved. (Contrastive loss uses Jensen-Shannon divergence, which yields log 2 constant and vanishing gradients for disjoint distributions of positive and negative sampled pairs (Zhu et al., 2021). Aggregations are less likely to be disjoint.

---

### Official Review · Reviewer_GN9c · 2022-10-27

**Confidence:** 4
**Clarity, Quality, Novelty And Reproducibility:** Overall, the propsoed method is novel…
**Correctness:** 3
**Technical Novelty And Significance:** 3
**Empirical Novelty And Significance:** 2
**Recommendation:** 3

**Strength And Weaknesses:**

Strenghts:
1. The motivation of this paper is good and solid. The computation complexity of InfoNCE loss is quadratic with respect to the number of nodes, which severly prevent full-graph training.
2. The proposed method does show better efficacy than traditional contrastive methods.

Weaknesses:
  1. The authors claim that contrastive learning on graphs require a large number fo negative samples while subsampling is suboptimal. Is there any empirical support for this claim as according to my experience, subsampling will not severly degrade the model’s performance.

2. Compared with data augmentation based contrastive learning  (InfoNCE loss) methods which ony requires two shared GNN encoder to generate node embeddings, the proposed method looks much more complicated (e.g., pretraining to get soft clusters). Although the overall complexity is linear, I doubt whether it can lead to efficacy when really applied to these datasets (especially when the graph is not that large, e.g., the datasets used in experiments)
3. The paper focus on contrastive learning on graphs, however, a lot of important related works are missing in both related works and experiments. For example, [1] and [2] are two data augmentation-based contrastive methods using InfoNCE loss. [3] avoids negative samples using asymmetric structures. [4] avoids negative samples through feature-level decorrelation. The complexity of [3] and [4] are both linear to the graph size and thus they are scalable. However, this paper never consider these important baselines.
4.  I am also confused about the tasks and datasets used in experiments. According to my knowledge, most self-supervised learning methods (including contrastive ones) foucs on node classifcation tasks (e.g., [1-4]). Why you consider graph clustering tasks instead of more commonly used node classication tasks.
5. Although the most imporant claimed advantage is scalability, the datasets used for evaluation are really small. The authors should consider use larger graphs.

References:
[1] Yanqiao Zhu, Yichen Xu, Feng Yu, Qiang Liu, Shu Wu, and Liang Wang. Deep graph contrastive representation learning. arXiv preprint arXiv:2006.04131, 2020b.
[2] Yanqiao Zhu, Yichen Xu, Feng Yu, Qiang Liu, Shu Wu, and Liang Wang. Graph contrastive learning with adaptive augmentation. In WWW, 2021.
[3] Shantanu Thakoor, Corentin Tallec, Mohammad Gheshlaghi Azar, Rémi Munos, Petar Velickovic, and Michal Valko. Bootstrapped representation learning on graphs. arXiv preprint arXiv:2102.06514, 2021.
[4] Hengrui Zhang, Qitian Wu, Junchi Yan, David Wipf, and Philip S Yu. From canonical correlation analysis to self-supervised graph neural networks. In NeurIPS, 2021.



**Summary Of The Paper:**

This paper proposes Proxy approximated meta-node Contrastive (PamC) for contrastive representation learning on graphs. PamC is motivated by the computational burden of vanilla contrastive loss (i.e., InfoNCE), and to deal with this problem, it proposes a meta-node based approximation technique which proxies all negative combinations in quadratic cluster size time complexity. Empirical reuslts sho that the proposed method demonstrates promising accuracy gains over sota graph clustering methods on 6 benchmarks with better efficiency.


**Summary Of The Review:**

Generally, I think the motivation of this paper is good. However, I think the propsoed method is over-complicted while does not show prominently better performance than simple methods. Besides, I believe the proposed method is not properly evaluated, in terms of tasks, datasets and baselines. I am leaning on rejection.

---

> ### Author Response · Authors · 2022-11-14
> **Response Q1 to Q2**
>
> **Q1: On contrastive learning on graphs require a large number fo negative samples**
>
> We claimed “Sub-sampling is not optimal and incorrect negative sampling leads to sampling bias.”
>
> [10] outlines core properties of a negative sampler for contrastive representation learning as efficiency, effectiveness, stability, and data independence, claiming that at present there is no method that meets all these requirements.
>
> [11] states that “Existing empirical work has established that larger number of negative samples consistently leads to better downstream task performances (Wu et al., 2018; Tian et al., 2019; He et al., 2019; Chen et al., 2020a), and often uses very large values (e.g., M = 65536 in He et al. (2019))”. We acknowledge that this includes InfoNCE type contrastive loss. However, the same noise contrastive estimation (NCE) technique is the core in most popular graph representation learning models.
>
> Additionally, any sub sampling approach, in a self-supervised scenario invariably results in incorrect negatives being chosen, (as class labels are unavailable), leading to (sub-optimal) sampling bias, further degrading performance.
>
> We avoid sampling by using the full node set, via the proxy. Our negative contrasting is efficient as it only requires additions; effective because we learn soft pseudo labels, stable and block contrastive and finally data independent. We use the same model on graphs, images, text and sensor data with no modifications.
>
> We have made the following changes in the paper:
>
> 1. In Abstract we reworded as "Sub-sampling is not optimal. Incorrect negative sampling leads to sampling bias”
>
> 2. In page 2, para 1, added: “However, while graphs particularly suited for contrastive learning, to be effective, a large number of negative samples must be used (Wang & Isola,2020) (e.g., 65536 in He et al. (2020)), along with larger batch sizes and longer training compared to its supervised counterparts (Chen et al., 2020b)
>
>
>
> **Q2. On comparative complexity against augmentation based contrastive learning (InfoNCE loss) methods**
>
> Our approach is to replace the negative sampling with an efficient proxy.
>
> A critical assumption underlying domain-agnostic graph augmentation is that only a fraction of the original graph is modified, thus task-relevant information is not significantly altered. It has been shown [12] that that this does not hold for many datasets. PAMC is not constraint by this assumption.
>
> 1. As a background to our motivation, we have added “However, all these works use some form of data augmentation which assumes that the task-relevant information is not significantly altered and require computationally expensive operations” in para 4 of page 2.
>
> The presence of a pre-train stage is not indicative of complexity. For example, consider two augmentation based methods: BGRL[4] and GRACE[1]. BGRL uses 4 encoder computations per update step (2 for target/online encoders, and 2 for each augmentation) plus a prediction step. GRACE does 2 (one for each augmentation), plus a projection step. Both methods backpropagate the learning signal 2 times (once for each augmentation), and we assume the backward pass to be approximately as costly as a forward pass [4]. Additionally, augmentations are typically done per epoch. Note that BGRL,  “first learn node representations in a fully unsupervised manner, and a linear model is then trained on top of these frozen embeddings without flowing any gradients back to the graph encoder network” [1].
>
> In PAMC, pre-training is minimal, 30 epochs, not requiring graph structure (adjacency) information. Also, only a subset of nodes is sufficient, as the goal is to establish initial approximate clusters. Thereafter, we use 1 forward and 1 backward pass. Further, in inference, PAMC does not require structural information for inference further lowering complexity and improving efficiency.
>
>  2. We have added node size vs training time and GPU memory usage graphs (Figure 3, Abstract Figures 8 and 9) and a comparison with and w/o PAMC (Abstract Table 5) to support the existing section on complexity with experimental evidence.

---

> ### Author Response · Authors · 2022-11-14
> **Response Q3, Q4, Q5**
>
> **Q3. On missing important related works**
>
> Our work focuses on obtaining an efficient approximation for the negative samples, as a means to improve the efficiency of traditional contrastive loss. As a result, we mention pioneering work on contrastive learning and discuss some works on augmentation-based methods. We did not include specific implementation of InfoNCE due to this reason. We also found BGRL[4] interesting but felt it not directly related to the proxy approach we are attempting to solve.
>
>  In regard to base lines, the following table shows the accuracy on cite dataset for comparison. Due to time, we have not re-implemented the models.
> | Model                               | Accuracy – CITE dataset |
> |-------------------------------------|-------------------------|
> | GRACE [1]         | 71.7 $\pm$ 0.6    |
> | GCA [2]           | 71.2 $\pm$ 0.2    |
> | BGRL [3] | 71.6 $\pm$ 0.4    |
> | CCA-SSG [4]   | 73.1 $\pm$ 0.3          |
> | **PAMC(Ours)**               | **73.3 $\pm$ 0.2** |
>
> 1. We have included the above table in the Appendix (Table 4)
> 2. We also add the following at the end of para 1 of page 2 :
> “”Prior work has used data augmentation-based contrastive methods Zhu et al. (2020; 2021b), negative samples using asymmetric structures Thakoor et al. (2021) or avoided negative samples altogether via feature-level decorrelation Zhang et al. (2021b). While Thakoor et al. (2021); Zhang et al. (2021b) address complexity and scalability, as seen in Appendix Table 4, their performance can be further improved.”
>
> **Q4. On selecting a clustering task for experiments**
>
> We used a clustering task following There is a wealth of self-supervision methods that focus on clustering tasks (AGCN,SDCN,DFCN, especially because we are also using the same datasets, which come from multiple domain including graphs, images and text). We believe once representations are learnt, which our PAMC also does, these can be used for any downstream task including clustering and classification.
>
> On a broader level, classification typically assigns exemplars to predefined categories. Clustering is typically about grouping related things (nodes) together on some notion of similarity. Importantly there is no predefined categories. After the clusters are chosen, assigning to existing clusters starts to look a lot like classification. However, as clusters typically seen as arbitrary, the main difference is in the evaluation methods. Instead of accuracy as in classification, clustering is evaluated subjectively in terms of usefulness for some particular tasks as well as the tendency to partition semantically. We see this aspect, and the ability to handle learn representation using unlabelled data, as a better and more generic approach when it comes to self-supervised representation generation.
>
> **Q5. On scalability and larger graphs**
>
> We acknowledge that we have not used larger graphs.
>
> We provide a through complexity analysis (Section 2.4) in the paper to address scalability theoretically.
> Additionally, we mention the benchmark dataset sizes to show the scalability benefits with and without PAMC in our paper.
>
> To address this concern now we have
> 1. Added a new Figure 8 to show the dataset node sizes with the time and memory gains. With PAMC approximation, there is a consistent reduction in training time and GPU memory usage positively correlated with the graph size. The efficiency benefits of the approximation increase with graph size, as expected theoretically.
> 2. Added a new Figures 3 and 6, showing the time and memory against node size in the synthetic graph and the following to the Performance section.
> “” Additionally, we used CITE dataset (3327 nodes) to create synthetics nodes. For a scale factor n, as contact nodes n times, along with edge-lists. Figure 3(right) shows the scaled edges and nodes for scale factors 5, 10, 15 · · · 45 and the GPU memory and training time for 1 epoch on Google colab T4 GPU with 16GB memory. Without PamC, scales over 5 is not possible due to running out of memory. With PamC over x45 (150,000 nodes) is possible. GPU and memory increase is liner confirming the theoretical time complexity. We used CITE as it is a very common dataset. We used synthetic node creation to capture variation over node size. Appendix A.8 shows GPU time breakup. Appendix A.6 shows the CITE dataset results with PamC when scaled from 1 . . . 20 in steps of 1.”””

---

### Author Response · Authors · 2022-11-14
**Correction on Inference times and changes in the new version**

Based on reviewer 6uyq, we found that we have made a mistake in reporting the SCGC* inference findings. We took the timing from pytoch profiler on Colab in August. The performance for PAMC was taken in September. Between the time, Colab has updated internally. Running SCGC* and PAMC now gives identical inference times. Accordingly, we have removed the inference Chart (Figure 3) and replaced it with a scalability chart that shows how the memory and training time varies with graph node and edge sizes, amended reference to inference times and added more graphs to show performance with increasing node and edge sizes using a synthetic version of the CITE dataset.

We have uploaded a revised paper, and, in the feedback, we mention what was changed. We also highlight in blue the changes done in the revised uploaded version of the paper, some of which address other reviewer comments and concerns as well.

We apologize for the time taken to respond as we were looking into explanations and updating the paper, and look forward to your responses. We also sincerely thank all the reviewers for the valuable comments, questions and engagement so far.

**Addressing novelty of the work and impact**

Our core novelty is the efficient proxy mechanism for the negative sampling. We show that it can work on graphs, images, text and (sensor) data. We also retrofit this on other state of the art models (AGCN, SDCN) and show that, adding PAMC is sufficient to improve these models.  Thus, our work can be seen as an enhancement to the broader contrastive learning technique, in terms of efficiency, robustness and accuracy, with broader implications.

In our work, we show that it is possible to replace the quadratic negative sampling process (Equation 1, denominator) by an approximation using meta-nodes, 𝜇^, shown in Equation 2.
We show this approximation has following benefits

1. Computation of the approximator, 𝜇^, only needs additions (a well optimized GPU primitive)
2. No sampling, hence, no sample bias.
3. time complexity drops from quadruple node to CxC, independent from node size and computable even on a CPU
4. Negative computation is no longer node based. There are no i,j terms on the right hand side. It is computed once per epoch.
5. it is block contrastive as we aggregate. Thus, it is robust and well behaved. (Contrastive loss uses Jensen-Shannon divergence, which yields log 2 constant and vanishing gradients for disjoint distributions of positive and negative sampled pairs (Zhu et al., 2021). Aggregations are less likely to be disjoint.

---

### Decision · Program_Chairs · 2023-01-20

**Decision:**

Reject

**Justification For Why Not Higher Score:**

Unanimous agreement that the comparison with existing art should be substantially improved.

**Justification For Why Not Lower Score:**

n/a

**Metareview: Summary, Strengths And Weaknesses:**

The paper provides a method to speed up contrastive learning, by using clusters to construct negative samples. Contrastive learning is a strong method yet is known to be a computational challenge, hence the goal of reducing its computational complexity is well motivated. The approach taken in the paper is agreed by the reviewers to be sensible,
and the experiments convincingly show the method reduces runtime without hurting performance when compared to baseline approaches for contrastive learning. The main problem with the paper, raised in some way in all of the reviews, is its comparison to existing works. The reviews raise several papers that this approach should be better positioned against.
There were some concerns about the techniques having limited novelty when compared against these previous works, but this could be an issue that can be resolved with a clearer comparison. The authors gave some explanations in the rebuttal phase, but the reviewers and I feel that the issue is too major to be fixed in a rebuttal phase.
The paper has potential, but requires a revision and another round of reviews before it can be accepted to a venue such as ICLR.